# Effectiveness evaluation of China's water resource tax reform pilot and path optimization from the perspective of policy field

**Mingyi Yang[1], Muqi Zhou[2], Conglin Zhang[3]***

1 School of Public Administration, Beihang University, Beijing, China, 2 School of Business, Hebei Normal University for Nationalities, Chengde, Hebei, China, 3 Institutes of Science and Development, Chinese Academy of Sciences, Beijing, China

* zhangconglin@casisd.cn

**Data Availability Statement:** All relevant data are within the manuscript and its Supporting information files.

## Abstract

The water resource tax reform played an important role in promoting sustainable development in China. Subsequent to the seven-year reform, the effectiveness evaluation of the policy in each pilot area and the exploration of the optimization path directly affected the promotion of water resource tax policy and the improvement of water use efficiency. Therefore, the theoretical framework of the water resource tax policy field was constructed to examine the mechanism of the three subsystems of policy scenario, policy orientation, and policy effect; fuzzy-set qualitative comparative analysis (fsQCA) was then used to evaluate and quantitatively compare the policy implementation effect and policy path in each pilot area, with emphasis put on three policy orientations, i.e., the decision and decomposition effect of policy goals, the selection and implementation effect of policy tools, and the policy supervision and security effect. As shown by the research results: ① the water resource tax reform had effectively improved the efficiency of water resource utilization in the pilot areas; ② three pilot models of water resource tax policy had been extracted, namely the policy goal and tool-driven model centering on a single dimension of the policy field, the implementation-supervision dual drive model emphasizing the supervision and security effect of the policy, and the three-dimensional policy orientation linkage model that focused on the synergistic effect of the policy field; ③ strong heterogeneity existed in water resource tax policy implementation paths and effects in each pilot area. Accordingly, regional heterogeneity could be considered in the process of reform to construct institutionalized, precise, and differentiated reform implementation methods from the perspective of the policy field.

## Introduction

Water resources are considered basic and strategic resources concerning people's livelihood. Rich in total water resources but low in per capita share, China is classified by the United Nations as a water-poor country [1]. Worse still, the low efficiency of water resource utilization

**Funding:** The sources of funding for our study are "Ministry of Water Resources of China" (Grant numbers[E203101901]). The funders had no role in study design, data collection and analysis, decision to publish, or preparation of the manuscript. None authors received a salary from any of our funders.

**Competing interests:** The authors have declared that no competing interests exist.

has become an important issue limiting the quality development of the Chinese economy and society. The key measures to promote the comprehensive conservation and recycling of resources and the formation of green development and lifestyle lie in conserving water resources and improving the efficiency of water resource utilization, which are also urgent requirements for *China's National Plan on Implementation of the 2030 Agenda for Sustainable Development* [2]. The levy of water resource fees is an important way to promote the paid use of water resources before the launch of the water resource tax pilot, but the problem of insufficient policy binding force and supervision capacity of the water resource fee system on the protection and utilization of water resources has become increasingly prominent. Therefore, following the remarkable results of the water resource tax pilot in Hebei Province in 2016 [3], nine pilot areas were designated in 2017 for the new water resource tax reform, including Beijing, Tianjin, Shanxi, Inner Mongolia, Henan, Shandong, Sichuan, Ningxia, and Shanxi [4]. As the intersection of the reform of the property rights system for natural resource assets and the reform of the compensation system for the paid use of resources and ecological protection, the water resource tax reform plays an important role in upgrading the construction of ecological civilization in China to a new level. After the seven years since the pilot was launched, has it effectively improved the efficiency of water resource utilization? What are the differences in the effectiveness of the pilot scheme in each pilot area? How can the water tax policy be further adjusted and improved? The answers to these questions are crucial to promoting China's overall water use efficiency, improving water use structure, building a water-saving society, and ensuring China's water security.

## Literature review

Studies do share some consensus that institutional factors must be considered if we want to effectively solve the problems of resource consumption and environmental pollution [5]. Research related to the evaluation and optimization of water resource policy has witnessed specific progress:

Some scholars have used case study analysis [6, 7], comparative analysis [7], and content analysis [8] to qualitatively analyze the policies related to water resource tax. Specifically, the practical experiences of water resources taxes in Italy, France, the European Union and other countries and regions were sorted out and compared [6, 7]. The reform status and effectiveness of various pilot areas of China's water resources tax reform were deeply explored [8, 9].

Computable general equilibrium model [8], (progressive) double difference method [10], general equilibrium model [11], synthetic control method [9], multiple regression model [10, 12, 13], and other research methods have been applied to quantitatively portray the implementation effect of water tax policy and its influencing factors. Specifically, the effectiveness of water resource tax reform was evaluated in terms of water use efficiency [10], reduction of negative resource externalities [9], amount of profits and taxes [8, 13]. In addition, the direct and indirect impacts of policy changes on all aspects of society are explored from the impact of water resources tax on water use, production and trade patterns, and the scale of welfare losses [11].

Existing studies have shown that the implementation effect of water resources tax is significantly affected by various factors, and there is spatial heterogeneity [10]. Technological progress [14], water resource endowment [15], regional economic development level [14], acreage and fertilizer use [16] all had a significant impact on the implementation effect of water resources tax.

Scholars believe that there are still many problems in the implementation of water resource tax policy, including the lack of targeted subsidies [17] and difficulties in balancing the differences in interests between different water-using subjects [18]. Scholars have put forward

corresponding countermeasures and suggestions. Specifically, the design of water resource tax rates should balance fiscal and economic objectives and that other policy tools are needed to help poor water users [17]. In addition, countermeasures have been proposed in terms of trade-offs between the economic or environmental values associated with water resources [19], increment in the flexibility of water resource management [20], and integration of theory and practice in the design of water resource tax [18].

Qualitative comparative analysis (QCA) is a research method between case-oriented (qualitative method) and variable-oriented (quantitative method), it is also a comprehensive research strategy that can take advantage of both methods [21]. According to the different set forms, the qualitative comparative analysis method can be subdivided into crisp set qualitative comparative analysis (csQCA), fuzzy set qualitative comparative analysis (fsQCA) and multi-value qualitative comparative analysis (mvQCA) [21]. Because this method can inductively analyze the nature of its configuration in a limited number of cases, it was widely employed in the fields of sociology, economics and management. In the field of resource and environmental policy, the fsQCA methodology has provided scholars with strong methodological support and developed rich results by utilizing its advantage of small-sample case comparison. Specifically, different categories of waste separation policies in 46 pilot cities in China were summarized and compared [22]. Besides, some scholars have applied fsQCA methods to understand how multiple sets of government policies, as captured by public expenditure on climate change issues (environment protection, pollution abatement, waste management) and fiscal strategies (energy, pollution, transport), provide positive or negative ecological footprints [23].

In general, the existing studies have witnessed much progress but still with room for further improvement: (1) there are relatively few existing studies that evaluate the effects of water resource tax policies from the perspective of public policy; (2) quantitative comparisons of implementation effectiveness of water resource tax in different pilot areas have rarely been made; (3) there are few related studies on the analysis of the influence paths of the factors affecting the implementation effectiveness of water resource tax.

The novelty of our research lies in combining the evaluation of the policy process and quantitative comparison of cases, taking the "1+9" water resource tax reform pilot areas in China as a case study from the perspective of regional heterogeneity, and carrying out the evaluation of the policy effect and quantitative comparative analysis of the policy path in a combination of the policy process, policy scenario, and policy field theory, thus providing support for adjusting and improving the water resource tax policy, enhancing water use efficiency and optimizing water use structure.

## Research methodology

Based on the water resource tax policy field, a quantitative comparative analysis method of policy paths is established. With the most representative regional heterogeneity factors and each evaluation index of the policy effect in the water resource tax policy scenario taken as the condition variables affecting the water resource tax policy effect, a logical connection based on the principle of set theory under the framework of configuration thinking is established between the configuration paths and results of multiple antecedent conditions to examine how multiple condition variables form a policy combination orientation and ultimately affect the policy effect.

### Research design

This article employs the fuzzy set qualitative comparative analysis method (fsQCA) to evaluate the policy effects of the "1+9" water resource tax reform pilots and compare the differences in the policy implementation effects of each pilot.

Compared with traditional symmetric (such as correlation analysis and multiple regression analysis) data analysis methods, Qualitative Comparative Analysis (QCA) developed by Charles Ragin is an asymmetric (individual case analysis) data analysis tool [24]. Combining case-oriented qualitative analysis with variable-oriented quantitative analysis [21] allows us to explore the complex interaction between two (or more) variables, that is, to explore how the configuration of conditional variables will affect the results [22], thereby producing more general analytical inferences and achieving both reasoning process and methodological rigor. This method has been widely used in the research of complex qualitative comparative problems in various fields.

Existing research on the demonstration effect of water resource tax reform mainly uses traditional statistical methods, focusing on the net effect or two-way interaction of outcome conditions, and exploring the relationship between water resource tax reform and water resource utilization efficiency and other variables. While water resources tax reform is a non-linear process involving multiple links such as policy design, policy implementation and policy supervision, fsQCA follows the set theory principle under the configuration thinking framework to explain how multiple conditions interact and establish a logical connection between the configuration path of multiple antecedent conditions and the result [21], which is suitable for the complex social issues caused by "multiple complex concurrent causation" in this study.

## Data source and case selection

In July 2016, Hebei Province became the first pilot area for water resources tax reform in China. In November 2017, the Ministry of Finance of China, China State Administration of Taxation, and the Ministry of Water Resources of China jointly issued the *Implementation Measures for Expanding the Pilot Program of Water Resources Tax Reform*, which expanded the scope of the pilot to 9 provinces (autonomous regions and municipalities) including Beijing, Tianjin, Shanxi, Inner Mongolia, Henan, Shandong, Sichuan, Ningxia and Shaanxi [4]. This study selects "1+9" water resources tax reform pilot areas as the main body of the case study, the location of these case study areas is shown in Fig 1.

The policy text comes from currently effective water resource tax-related measures, plans, opinions, etc., such as the "Interim Measures for Water Resources Tax", "Measures for Expanding the Implementation of Water Resources Tax", and the "Measures for the Implementation of Water Resources Tax" of various provinces (cities and districts). The provincial government portal website, official website of functional departments, Peking University Magic Database and other websites were searched, and data were collected with the help of the National Bureau of Statistics website and water resources bulletin.

## Theoretical framework

Both the rationalism perspective of policy process concerning policy formulation-implementation-supervision and the constructivism perspective of policy logic concerning policy goals-policy tools-policy actors (stakeholders) are traditional dimensions of policy evaluation, but neither of them can effectively reconcile the contradictions between "fact" and "value" evaluation [25, 26]. And promoting the mutual complementarity of rationalism and constructivism proves to be better in optimizing the evaluation methods of public policies and upgrading public policy evaluation to be more scientific, objective, fair, and professional. In order to achieve a balance between value and fact, process and result, logic and effect, static and dynamic, and other aspects of policy evaluation, the water resource tax policy field is constructed to examine the operation mechanism among the three subsystems of policy scenario, policy orientation, and policy effect, and to realize the combination of the dynamic operation process and static

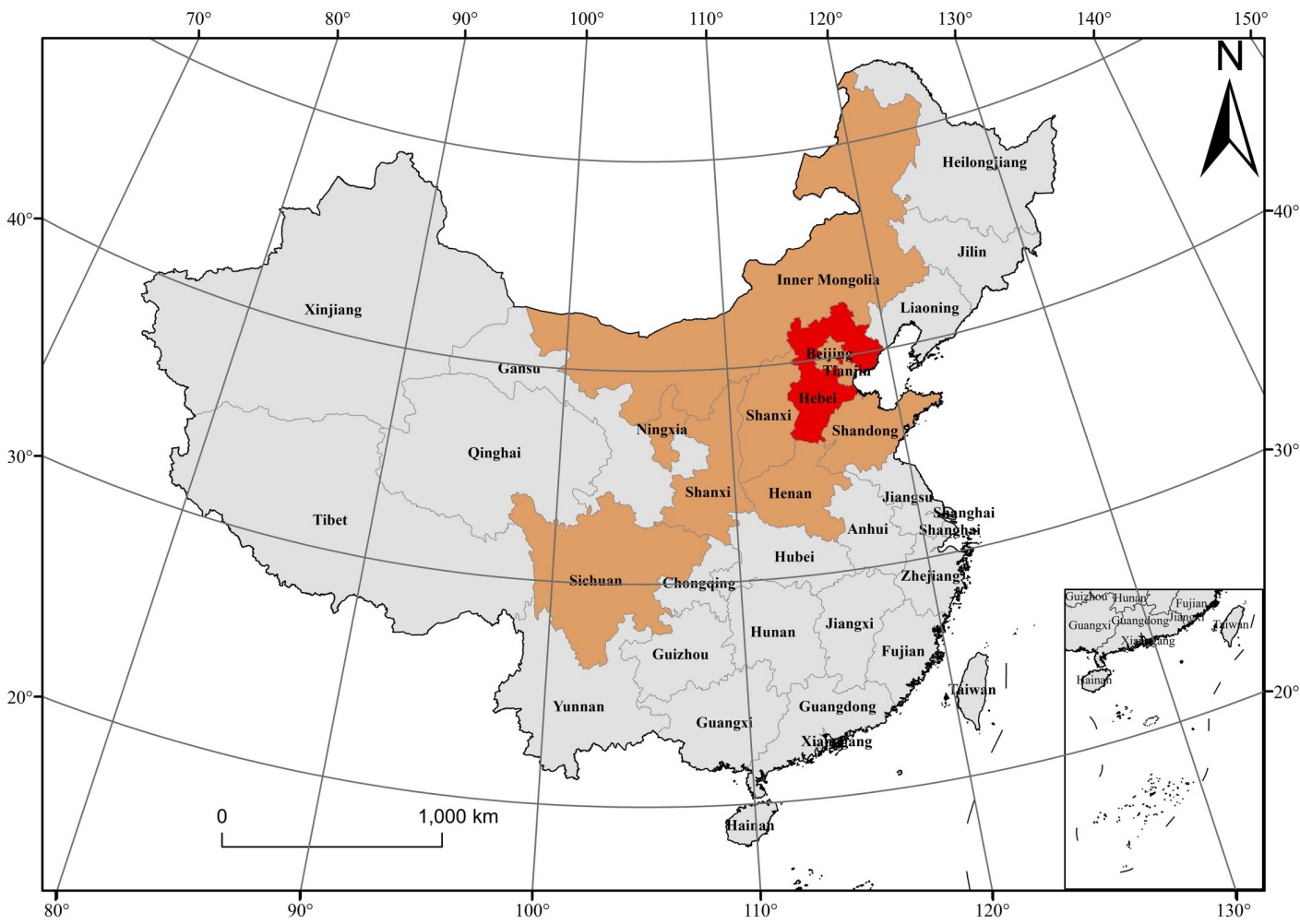

**Fig 1. The location of "1+9" water resources tax reform pilot areas.**

logic structure. In other words, it is suggestive to fully consider how the interaction between the actors and the policy objects in the water resource tax policy field in a regionally heterogeneous policy scenario forms the policy orientation and ultimately produces the policy effect, which provides a theoretical basis for the subsequent evaluation of the policy effect and quantification and comparison of the policy path (Fig 2).

**Policy scenario.**   Policy scenario is characterized by regional heterogeneity in space and dynamic changes in time [27, 28]. While the public-good attribute of natural resources requires systemic integrity to be considered in the formulation and implementation of resource policies, it is difficult to achieve an effective response to systemic policy problems with a single policy tool or a simple superposition of multiple policy tools in traditional policy practice [29]. Therefore, for resource policies in complex scenarios, it is more important to focus on the interaction between the policy itself and the external environment and conditions and consider the relationship between heterogeneous policy scenarios and policy orientations and their combinations comprehensively. Studies have shown that water resource endowment is an important factor affecting regional water use efficiency, and there is often a negative correlation between the abundance of water resources and the awareness of water conservation

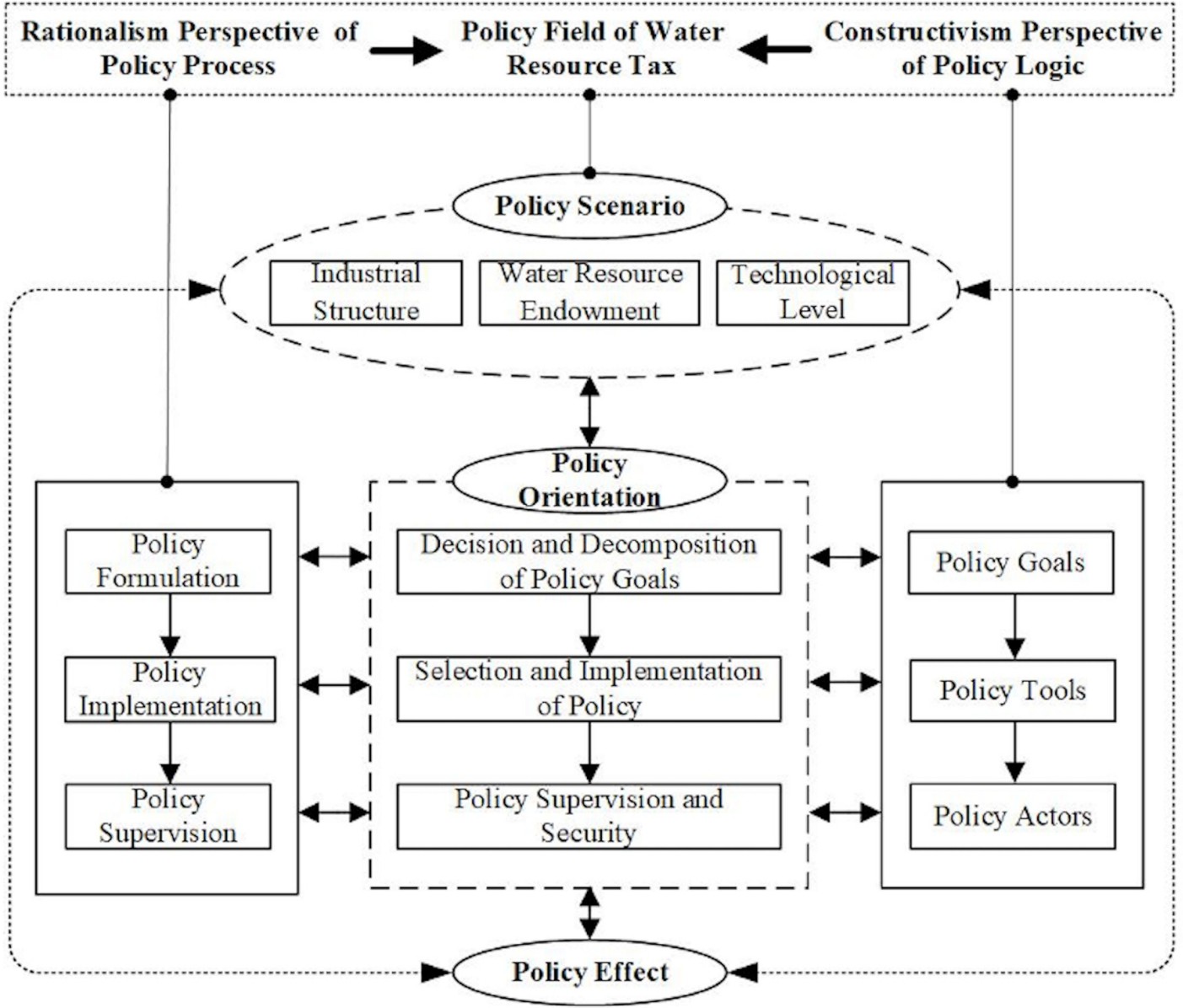

**Fig 2. Theoretical framework of water resource tax policy field.**

among residents in the region, which directly affects water use efficiency; industrial structure constitutes a key factor influencing water use efficiency in different regions, and large differences exist in water use efficiency among various types of industries; regional technological level can, to a certain extent, reflect the elimination of outdated high water-consuming processes and related equipment and the development of efficient water-saving technology, which plays a positive role in improving water use efficiency [30]. Therefore, the subsystem of water resource tax policy in this paper is mainly portrayed by three regional heterogeneity factors, namely water resource endowment, industrial structure, and technological level.

**Policy orientation.** Following the systemic principle of the policy field, the rationalism perspective of the policy process and the constructivism perspective of policy logic are

combined to summarize the policy orientation of water resource tax and extract the key elements from three levels with full consideration of the regional heterogeneous policy scenario. Specifically, under the policy orientation concerning the decision and decomposition of policy goals, the tax objects, tax rate design, and tax-free situation design are respectively the basic elements, core aspects, and important manifestations of the policy goals of tax system construction [31, 32]. Under the policy orientation of the selection and implementation of policy tools, the perfect tax collection and management model serves as an important guarantee for policy implementation, and the degree of tax collection in place is an important reflection of the ability to implement the policy in pilot areas [33]. The policy supervision and security orientation formed for the main body of water resource tax policy ensures the continuity and effectiveness of policy implementation, which is an important guarantee to achieve the policy goals. Among them, the enforcement and supervision of policy actors and the construction of assessment and accountability mechanisms constitute the core elements of building an effective policy supervision system [34].

**Policy effect.** The evaluation of water resource tax policies in the policy field should not only focus on the result orientation, but should also be oriented to the whole process and systemic integrity of the policy orientation. In specific policy scenarios, different policy orientations are often formulated according to local conditions, forming a policy configuration path with regional heterogeneity. Therefore, policy evaluation should emphasize the effect of the combination of policy orientations and the field effect of its formation under the differentiated policy scenario [35].

## Variable selection and assignment calibration

**Result variables.** Concerning existing studies, this paper selects regional water use efficiency as the result variable to represent the effect of the water resource tax reform in the policy field and uses the "annual average rate of change in water use per 10,000-yuan GDP" as an indicator to measure the change in water use efficiency in the pilot areas [36].

**Condition variables.** Three regional heterogeneity factors under "policy scenario" and seven key factors affecting the effectiveness of water resource tax policy under "policy orientation," namely "decision and decomposition of policy goals," "selection and implementation of policy tools," and "policy supervision and security," are selected as the condition variables for the quantitative comparison of water resource tax policy paths. On this basis, the condition variables are divided into five levels according to their performance to obtain the evaluation set $V = \{v_1, v_2, v_3, v_4, v_5\}$ = {Excellent, good, average, to be improved, urgent to be improved}; then, each level is assigned a value to obtain the evaluation score set $U = \{u_1, u_2, u_3, u_4, u_5\}$ = {1, 0.8, 0.6, 0.4, 0.2} [37].

**Variable calibration.** The numerical variables are calibrated by the direct calibration method by which 95%, 50%, and 5% quantile values are set as anchor points to calibrate the implementation effect of water resource tax from "A," "B," to "C" levels. The variable measurements are then converted to fuzzy scores from 0–1 by the fsQCA algorithm. For other types of variables, the indirect calibration method is adopted, and a 5-value assignment scheme is used according to the requirements of the fsQCA algorithm for the variable assignment [31]. The selection and calibration of the condition variables are shown in Table 1.

## Testing of necessary conditions and configuration conditions

**Necessary conditions.** The consistency measure is used to determine whether a single condition variable is necessary to influence the effect of water resource tax policy, and the explanatory strength of condition $X$ to $Y$ is determined by the coverage index:

**Table 1. Selection and calibration of condition variables.**

| Policy System | Evaluation Dimension | Variables | Evaluation Index and Assignment Descriptions | Reference | Data Sources |
|---|---|---|---|---|---|
| Policy Scenario ($A_1$) | Regional Heterogeneity Factor ($B_1$) | Industrial Structure ($C_1$) | The ratio of the tertiary industry to GDP in the pilot areas. The 95%, 50%, and 5% quartiles are set as anchor points for calibration. A ratio lower than 5% indicates a low level in the transformation and upgrading of the pilot industrial structure, while that higher than 95% represents a high level in the transformation and upgrading of the pilot industrial structure. | [15, 38] | National Bureau of Statistics |
| | | Water Resource Endowment ($C_2$) | Total actual year-end water resources in the pilot areas. The 95%, 50%, and 5% quartiles are set as anchor points for calibration. A ratio lower than 5% indicates an insufficiency in the pilot water resource endowment, while that higher than 95% represents the abundance in the pilot water resource endowment. | [39] | |
| | | Technological Level ($C_3$) | Expenditure on research and development in the pilot areas. The 95%, 50%, and 5% quartiles are set as anchor points for calibration. A ratio lower than 5% indicates a low level of technological progress in the pilot areas, while that higher than 95% represents a high level of technological progress in the pilot areas. | [15] | |
| Policy Orientation ($A_2$) | Decision and Decomposition of Policy Goals ($B_2$) | Tax Object Design ($C_4$) | ① If the "water resource tax rate table" integrates four types of tax items: surface water, groundwater, urban public water supply, and miscellaneous water use, the tax items set can be deemed complete. ② If the "miscellaneous water use" tax item of the "water resource tax rate table" sets sub-items of "ground-source heat pump use" and "thermal power generation cross-flow cooling water," then the specific sub-item set can be deemed complete. A value of 1 is assigned when the completion is both satisfied; a value of 0.8 is assigned with only ② is completed; a value of 0.6 is assigned with only ① is completed; a value of 0.2 is assigned when neither ① and ② are completed. | [40] | Each Pilot Area <Implementation Approach of Water Resource Tax Reform Pilot> |
| | | Tax Rate Design ($C_5$) | ① If differential tax rates are developed respectively for different industries, such as domestic water, agricultural water, industrial water, and special industry water, and for different uses, including hydroelectric power generation, thermal power generation, drainage, geopotential heat pumps, then the design of the differential tax rate standards can be considered complete. ② If heavy taxation is expressly levied on the overdrawing areas and seriously overdrawing areas that take groundwater, for example, with groundwater taken in the urban public water supply network coverage area, more tax is levied than the area not covered by the urban public water supply network, then the design of heavy taxation can be considered complete. ③ If the standard of tax reassessment for excess water withdrawal is less than 20% (inclusive), 2 times; 20%-30% (inclusive), 2.5 times; more than 30%, 3 times, then the design can be considered excellent; If less than 20% (inclusive), 2 times; 20%-40% (inclusive), 2.5 times; more than 40%, 3 times, then the design can be considered good; other standards are considered poor. With the design of ① and ② completed and ③ considered excellent, a value of 1 is assigned; with one of ① and ② uncompleted and ③ considered excellent, a value of 0.8 is assigned; with one of ① and ② uncompleted and ③ considered good, a value of 0.6 is assigned; with one of ① and ② uncompleted and ③; considered poor, a value of 0.4 is assigned; with both ① and ② uncompleted, a value of 0.2 is assigned. | [13] | |
| | | Tax-free Situation Design ($C_6$) | ① If the tax-free scope of sewage treatment water extends to reuse water, reclaimed water, rainwater, brackish groundwater, brackish water, desalinated seawater, and other non-conventional water sources, then the scope of tax exemption from sewage treatment water is reasonable; if the tax-free situation is only limited to the treatment of "reclaimed water" with sewage, then the scope is unreasonable. ② If the reinjection of oil drainage into the closed pipeline after separation and purification is involved in the tax-free category, the tax-free situation design is complete. With ① defined reasonably, a value of 1 is assigned; with ① defined unreasonably but ② designed completely, a value of 0.8 is assigned; with ① defined unreasonably and ② incomplete, a value of 0.2 is assigned. | [32, 40] | |
| | Selection and Implementation of Policy Tools ($B_3$) | Tax Collection and Management Model ($C_7$) | ① If a collaboration mechanism is established between the taxation and water conservancy departments, then the inter-departmental coordination and cooperation mechanism is sound. ② Concerning water loss in water supply enterprises, if a reasonable loss rate can be set according to the actual situation to calculate the water intake of water supply enterprises, then the taxation basis is considered reasonable. ③ If innovation is included in the levy model, including the measurement model, platform, tax model, and other aspects, then the innovation of the management model can be considered to be achieved. With ① considered sound, ② reasonable, and ③ achieving innovation, a value of 1 is assigned; with ① considered sound, ② reasonable, but ③ failing to achieve innovation, a value of 0.8 is assigned; with ① considered sound, ③ achieving innovation but ② unreasonable, a value of 0.6 is assigned; with ① considered sound, ② unreasonable and ③ failing to achieve innovation, a value of 0.2 is assigned. | [33] | Each Pilot Area <Implementation Approach of Water Resource Tax Reform Pilot> <Management Approach of Water Resource Tax Collection> |
| | | Tax Collection Situation ($C_8$) | ① The proportion of groundwater water tax collected by the tax authorities to the overall water tax after the tax reform. The proportion of each pilot area was ranked and calibrated using the 95%, 50%, and 5% quartiles as anchor points. A proportion below 5% indicates that the pilot water resource tax collection is not in place, while that above 95% means a favorable situation in the pilot water resource tax collection. | [40] | China Tax Yearbook |
| | | Regional Water-saving Effect ($C_9$) | The average annual change rate of groundwater withdrawal in the pilot areas. The change rates of each pilot area were ranked and calibrated using the 95%, 50%, and 5% quartiles as anchor points. A rate lower than 5% indicates favorable water-saving effects in the pilot area after the tax reform, while that higher than 95% warns of poor water-saving effects. | [16] | National Bureau of Statistics |
| | Policy Supervision and Security ($B_4$) | Supervision of Water Resource Tax ($C_{10}$) | ①If the public security, judicial and procuratorial departments increase the efforts to protect the tax, strictly enforce the water resource tax, and maintain a fair and unified market order, it is considered to be able to reflect enhanced law enforcement; otherwise, it is considered not to reflect enhanced law enforcement. ② If the pilot water resource tax reform is included in the assessment of relevant departments at all levels of government and holds the units accountable for ineffective law enforcement, the assessment and accountability mechanism is considered sound; otherwise, it is considered unsound. With both ① and ② considered sound, a value of 1 is assigned; with only one of them sound, a value of 0.8 is assigned; with both unsound, a value of 0.2 is assigned. | [40] | Each Pilot Area <Implementation Approach of Water Resource Tax Reform Pilot> |

Consistency Measure:

$$Consistency(X_i \leq Y_i) = \sum [min(X_i, Y_i)] / \sum X_i \qquad (1)$$

Coverage Measure:

$$Coverage(X_i \leq Y_i) = \sum [min(X_i, Y_i)] / \sum Y_i \qquad (2)$$

Among them, *Consistency* refers to the consistency, whose minimum threshold is 0.9, and above 0.9 is judged as a necessary condition [41]; *Coverage* is the coverage rate, and the higher the coverage rate, the better the explanatory power ability of condition $X$ to $Y$; $X_i$ denotes the affiliation of the $i$ pilot province in the condition combination $X$; $Y_i$ denotes the affiliation in the result $Y$ of the $i$ pilot province.

**Configuration conditions.** Conduct a conditional configuration adequacy analysis on multiple conditional variables that affect the effect of water resources tax policy to explore the multiple concurrent factors and complex causal mechanisms that affect policy implementation effects in the field of water resource tax policy effects. By quantitatively comparing the differences in policy implementation effects of the "1+9" water resource tax reform pilot, we can find a configuration path to optimize policy effects.

## Results

### Evaluation of implementation effect of water resource tax policy

The policy evaluation index scores of the "1+9" water resource tax reform pilot is shown in Table 2. The overall performance of the reform pilot areas is good regarding the decision and decomposition of policy goals. In some provinces, the local water resource characteristics can be combined with economic development structure to scientifically design water resource tax items, tax rates, and other tax system elements. Beijing, Shanxi, Inner Mongolia, and Henan belong to those provinces boasting relatively complete water resource tax items. For example, in each pilot area, "ground-source heat pump use" has been set up under "miscellaneous water use" of the "water resource tax rate table." However, there is no such sub-category in the "water resource tax rate table" of Sichuan. The tax rate design in Henan and Shandong is relatively scientific and reasonable, and the design of tax-free situations in most provinces is reasonably defined. In contrast, Beijing has failed to give full play to the incentive and constraint effects of water resource tax when designing tax rates and tax-free situations.

**Table 2. "1+9" water resource tax reform pilot policy evaluation index scores.**

| Policy System | Evaluation Dimension | Variables | Bei jing | Tian jin | Hebei | Shan xi | Inner Mongolia | Shan dong | He nan | Si chuan | Shaanxi | Ning xia |
|---|---|---|---|---|---|---|---|---|---|---|---|---|
| A₁ | B₁ | $C_1$ | 0.98 | 0.83 | 0.51 | 0.45 | 0.24 | 0.55 | 0.1 | 0.54 | 0.03 | 0.29 |
| | | $C_2$ | 0.12 | 0.04 | 0.44 | 0.4 | 0.83 | 0.61 | 0.57 | 0.98 | 0.85 | 0.06 |
| | | $C_3$ | 0.55 | 0.39 | 0.78 | 0.2 | 0.15 | 0.98 | 0.89 | 0.72 | 0.45 | 0.02 |
| A₂ | B₂ | $C_4$ | 1 | 0.8 | 0.8 | 1 | 1 | 0.8 | 1 | 0.2 | 0.6 | 0.6 |
| | | $C_5$ | 0.2 | 0.6 | 0.6 | 0.6 | 0.4 | 0.8 | 1 | 0.6 | 0.4 | 0.4 |
| | | $C_6$ | 0.2 | 0.8 | 1 | 0.8 | 0.8 | 0.8 | 0.8 | 0.8 | 0.8 | 0.8 |
| | B₃ | $C_7$ | 0.6 | 0.6 | 1 | 0.2 | 0.2 | 1 | 0.2 | 0.6 | 0.8 | 0.6 |
| | | $C_8$ | 0.12 | 0.1 | 0.94 | 0.96 | 0.65 | 0.75 | 0.37 | 0.02 | 0.73 | 0.27 |
| | | $C_9$ | 0.28 | 0.26 | 0.77 | 0.82 | 0.56 | 0.61 | 0.44 | 0.08 | 0.6 | 0.39 |
| | B₄ | $C_{10}$ | 0.6 | 0.4 | 1 | 0.6 | 0.2 | 0.8 | 0.2 | 0.2 | 0.2 | 0.2 |

Regarding the selection and implementation of policy tools, the tax collection and management model differs greatly among the pilot areas. Each pilot has established a collaborative mechanism between taxation and water conservancy departments, of which Hebei, Shandong, and Shaanxi deducted reasonable losses in calculating the actual water withdrawal and set a more reasonable taxation basis; Hebei and Shandong achieved innovation in the tax collection and management related to the measurement model of electricity and water and the use of the new media platform for government affairs; while Shanxi, Henan, and Inner Mongolia failed to deduct reasonable losses in calculating the actual water withdrawal, and did not achieve innovation in the tax collection and management model yet. In addition, from the perspective of the proportion of groundwater resource tax, the situation of water resource tax collection proved to be better in Hebei and Shanxi provinces than that in other provinces.

Regarding policy supervision and security, the overall policy supervision and security mechanism of the "1+9" pilot areas needs to be further improved. Among them, Hebei Province, as the first pilot area of the reform, has a sound policy assessment and accountability mechanism, and the synergy and linkage between taxation and water conservancy departments are strong. However, most pilot areas necessitate an improvement in law enforcement to investigate and deal with unauthorized drilling, forcible water withdrawal, delinquencies, and other behaviors.

## Construction of truth tables and analysis results of necessary conditions

A truth table is constructed (Table 3), in which each row represents a logically possible configuration, or combination of factors/conditions associated with a given outcome [37]. The presence (1) or absence (0) of conditions is specified for each row.

Table 4 reports results of our necessity analysis. No condition met or exceeded the consistency score threshold of 0.90, indicating that none of them can be explained as a necessary condition affecting the effect of water resource tax policy, and the influence of each condition variable on the policy effect has strong interdependence, and different condition combinations have different impacts on the policy effect. On this basis, the condition combinations are further tested and analyzed.

## Analysis results of configuration conditions

The configuration analysis is used to reveal which configuration paths of condition variables are sufficient conditions to influence the effect of water resource tax policy. Concerning threshold setting, the original consistency threshold should be set to ensure the equality of

**Table 3. Truth table.**

| Case | $C_1$ | $C_2$ | $C_3$ | $C_4$ | $C_5$ | $C_6$ | $C_7$ | $C_8$ | $C_9$ | $C_{10}$ |
|---|---|---|---|---|---|---|---|---|---|---|
| Beijing | 1 | 1 | 1 | 1 | 1 | 1 | 1 | 1 | 1 | 1 |
| Tianjin | 0 | 1 | 0 | 1 | 1 | 1 | 0 | 0 | 0 | 0 |
| Hebei | 1 | 0 | 0 | 1 | 0 | 0 | 1 | 0 | 0 | 0 |
| Shanxi | 1 | 1 | 1 | 0 | 1 | 1 | 1 | 0 | 0 | 1 |
| Inner Mongolia | 1 | 0 | 1 | 1 | 1 | 1 | 1 | 0 | 0 | 1 |
| Shandong | 0 | 0 | 0 | 1 | 1 | 1 | 1 | 1 | 1 | 0 |
| Henan | 0 | 1 | 0 | 1 | 0 | 1 | 1 | 0 | 1 | 0 |
| Sichua | 1 | 0 | 0 | 1 | 0 | 1 | 0 | 1 | 1 | 0 |
| Shaanxi | 1 | 1 | 1 | 0 | 1 | 1 | 0 | 0 | 0 | 1 |
| Ningxia | 0 | 0 | 0 | 1 | 1 | 1 | 0 | 1 | 1 | 0 |

**Table 4. Analysis of conditions for necessity consistency and coverage.**

| Condition | Necessity Consistency | Necessity Coverage |
|---|---|---|
| Water Resource Endowment | 0.73 | 0.87 |
| Industrial Structure | 0.54 | 0.84 |
| Technological Level | 0.57 | 0.81 |
| Tax Object Design | 0.75 | 0.84 |
| Tax Rate Design | 0.72 | 0.84 |
| Tax-free Situation Design | 0.67 | 0.81 |
| Tax Collection and Management Model | 0.57 | 0.81 |
| Tax Collection Situation | 0.79 | 0.86 |
| Regional Water-saving Effect | 0.71 | 0.87 |
| Supervision of Water Resource Tax | 0.69 | 0.80 |

rows of the truth table with results of 0 and 1. According to the data characteristics of the "1+9" case sample in the setting of condition variables and result variables, the frequency threshold is set to 1, and the original consistency threshold is set to 0.75. The intermediate solution with moderate complexity and strong rationality is selected as the analysis result from the three output solutions, and the core conditions and secondary conditions are distinguished by the simple solution (Table 5).

According to the arithmetic results (Table 5), there are four configuration paths affecting the effect of water tax policy, i.e., four different combinations of condition variables. The consistency of both the individual and overall solutions is higher than 0.75, which meets the acceptable criteria [42]. The overall solution consistency of this result is 0.91, and the overall solution coverage is 0.55, implying that the policy implementation yielded effects in about 91% of the pilot areas in all cases that satisfy the above four configuration paths. As the above four

**Table 5. Condition combinations for the effect of water resource tax policy.**

| Condition Variables | Condition Configuration Path | | | |
|---|---|---|---|---|
| | H1 | H2 | H3 | H4 |
| Water Resource Endowment | ☆ | ● | ☆ | ● |
| Industrial Structure | ☆ | ☆ | | ○ |
| Technological Level | ☆ | ○ | ☆ | ○ |
| Tax Object Design | ○ | ○ | ○ | ☆ |
| Tax Rate Design | ● | ○ | ○ | ● |
| Tax-free Situation Design | ○ | ○ | ○ | ○ |
| Tax Collection and Management Model | ☆ | ● | ● | ● |
| Tax Collection Situation | ○ | ☆ | ○ | ☆ |
| Regional Water-saving Effect | ○ | ☆ | ○ | ☆ |
| Supervision of Water Resource Tax | ☆ | ● | ☆ | ○ |
| Consistency | 0.92 | 1.00 | 1.00 | 0.83 |
| Original Coverage | 0.25 | 0.22 | 0.14 | 0.14 |
| Unique Coverage | 0.13 | 0.10 | 0.06 | 0.12 |
| Overall Solution Consistency | 0.91 | | | |
| Overall Solution Coverage | 0.55 | | | |

Note: ● means that the core condition exists, ○ the secondary condition exists, ★ the core condition does not exist, and ☆ the secondary condition does not exist.

configuration paths together explain 55% of the cases, it embodies a high degree of explanation overall.

## Discussion

Overall, the four paths can represent four policy combination orientations and patterns of water resource tax reform pilot areas in regionally heterogeneous policy scenarios, represented by the provinces of Shanxi, Shaanxi, Henan, and Sichuan. In terms of the core conditions of all paths, water resource endowment, tax rate design, and tax collection and management model appear as core conditions in at least two paths, indicating that these three factors play a key role in the water resource policy field and have the most significant impact on the policy effects. Based on the empirical results, the combination of water resource tax policy orientations in each region can be categorized into the following three patterns.

(1) Configuration Path 1 and Configuration Path 3: Policy Goal and Tool-driven Models

The original coverage rate of configuration path 1 is 0.25, and the unique coverage rate is 0.13, which means that this path can explain about 25% of the cases, and 13% of them can be explained by this path only. In path 1 (H1), "tax rate design" appears in the form of core conditions, "tax object design," "tax-free situation design," "tax collection situation," "tax collection situation," and "regional water-saving effect" appear as secondary conditions, and the regional heterogeneity factor, as a secondary condition, does not exist.

The original coverage of path 3 is 0.14, and the unique coverage is 0.06, which indicates that this path can explain about 14% of the cases, of which 6% of the cases can be explained by this path only. In this path, the "tax collection and management model" appears as the core condition, "tax rate design," "tax object design," "tax-free situation design," "tax collection situation," and "regional water-saving effect" appear as the form of secondary conditions.

It can be concluded from the common features of these two paths that in non-specific regional policy scenarios, the improvement in the decision and decomposition functions in the water resource tax policy field supplemented by the enhancement in the policy tool selection and implementation functions, especially focusing on improving the tax rate design and tax collection model, can help improve the overall effect of water resource tax policy. In both paths, regional heterogeneity factors such as "water resource endowment," "industrial structure," and "technological level" are not the core conditions affecting the policy effect. As the policy supervision and security policy orientation of the pilot areas covered by these two paths is not that obvious, this model is named a policy goal and tool-driven model.

The representative province of this model is Shanxi Province. As a major water user, Shanxi is a province with average-level water use efficiency among the ten pilot areas. At this stage, the policy effects are mainly consolidated by strengthening the selection and implementation process of policy tools. In order to support the local economic development by consuming more water resources, Shanxi Province, when designing the tax rate, was able to set different tax rates for different water-using industries and give certain preferences in the tax rate range based on the realities of local water resources. Thus, Shanxi Province can flexibly and scientifically leverage water resource tax in water resource policy design to support the macro policy goal of "strengthening water resource management and protection, and promoting the conservation and rational development and utilization of water resources." At the same time, among the "1+9" pilot areas, the taxation department of Shanxi Province collected the highest proportion of groundwater water tax after the tax reform with favorable conditions in tax collection, highlighting the supreme capability of Shanxi Province in water resource tax policy implementation.

(2) Configuration Path 2: Implementation-supervision Dual Drive Model

The original coverage rate of this path is 0.22, and the unique coverage rate is 0.10, indicating that this path can explain 22% of the cases, and 10% of them can only be explained by this path. In the configuration path 2 (H2), "tax collection and management model" and "supervision of water resource tax" appear as core conditions, and "tax object design," "tax rate design," and "tax-free situation design" exist as secondary conditions, and the regional heterogeneity factor of water resource endowment appears as a core condition, while "technological level" exists as a secondary condition. This suggests that the water resource endowment and technological level of the regions covered by this path exert a strong influence on the effectiveness of water resource tax policy; meanwhile, these regions attach great importance to the selection and implementation of policy tools and policy supervision and security, especially the construction and design of tax collection and management model. Therefore, this model can be summarized as an implementation-supervision dual-drive model.

The representative province of this path is Shaanxi, a province sharing a high water endowment and water use efficiency among the ten pilot areas. Therefore, in the process of tax reform, Shaanxi Province has established a collaborative tax collection mechanism between local taxation and water administration authorities. The water administration authorities regularly send relevant water resource management information, such as the actual water consumption of water-using companies and individuals, to the local taxation authorities, and taxpayers declare the taxes to the local taxation authorities based on the information such as the actual water consumption approved by the water administration authorities, and then the local taxation authorities collect water resource tax and send information regularly to administration authorities. What's more, Shaanxi Province is able to deduct the reasonable losses of water supply enterprises at a 12% loss rate when calculating the actual water consumption. All these measures have intensified the scientific and rational nature of the water tax collection process to a certain extent.

(3) Configuration Path 4: Three-dimensional Policy Orientation Linkage Model

The original coverage rate of this configuration path is 0.14, and the unique coverage rate is 0.12, which means that this path can explain about 14% of the cases, and 12% of the cases can be explained by this path only. In configuration path 4 (H4), "tax rate design" and "tax collection and management model" appear as core conditions, and "tax-free situation design" and "supervision of water resource tax" appear as secondary conditions; among the regional heterogeneity factors, water resource endowment appears as a core condition, while "industrial structure" and "technological level" exist as secondary conditions. This path can be interpreted that the three-dimensional policy orientation linkage of the decision and decomposition of policy goals, the selection and implementation of policy tools, and policy supervision and security will directly strengthen the policy effect in a policy scenario with rich water resource endowment, a better industrial structure, and high technological level. In this way, this path can be named the three-dimensional policy orientation linkage model.

The representative province of this model is Sichuan, a province with high water resource endowment and relatively high water use efficiency among the "1+9" pilot areas. The design of tax rates and tax-free situations is relatively complete, and a collaborative mechanism between taxation and water conservancy departments has been set up. Innovation has also been realized in the water resource tax collection and management model. With the combination of big data and collection management and innovation in the non-contact tax model, its tax collection and management model is relatively sound, which generally helps Sichuan Province improve water use efficiency and achieve better water-saving effects in the process of tax reform.

## Conclusion

This paper constructs a general theoretical and methodological framework for public policy evaluation, which is "the decision and decomposition of policy goals—the selection and implementation of policy goals—policy supervision and security." In addition, public policy evaluation and natural resource measurement methods are combined to construct an evaluation index system from the perspective of the water resource tax policy field to evaluate the effect of the policy, and then conduct a quantitative comparative analysis of the policy path of "1+9" water resource tax pilot areas from the perspective of regional heterogeneity.

The analysis establishes two things. The core conditions of all the configuration paths include "water resource endowment," "tax rate design," and "tax collection situation," proving that there is a strong interaction between policy scenario, policy orientation, and policy effect. Therefore, with an in-depth analysis of the policy scenario in the policy field, the enrichment and adjustment in the combination of policy orientations can yield better policy effects; at the same time, by improving the heterogeneity factors that can be artificially regulated in the policy scenario, the overall effect of the policy field can be improved. The typical representative regions of the three configuration paths, Shanxi, Shaanxi, and Sichuan, differ greatly in terms of water resource endowment, industrial structure, and technological level, indicating strong heterogeneity in the implementation paths and effects of water resource tax policies in these pilot areas.

The main contribution of this research is to establish a set of resource tax policy evaluation and comparison methods combining policy process evaluation and quantitative comparison of cases. Meanwhile, it provides reference for China's water resources tax policy adjustment and subsequent expansion of the pilot scope and national promotion. Compared with other studies on water resources tax evaluation, our research does not directly measure the efficiency of water resources tax, but evaluates and compares the water resources tax reforms in each pilot region in China based on the policy field perspective.

This analysis needs to be extended in several ways and a number of limitations apply. Firstly, water resources tax reform is regarded as a holistic policy in this study, but in the real situation the effect of water resources tax reform may be affected by the industry in which it is located. Secondly, regional water use efficiency is selected as the outcome variable in this study, but in reality, the evaluation dimensions and indicators of water resources tax reform effects are diversified. All this is deferred to future research.

## Supporting information

**S1 Data.**
(CSV)

## Author Contributions

**Conceptualization:** Conglin Zhang.

**Data curation:** Muqi Zhou.

**Formal analysis:** Mingyi Yang.

**Investigation:** Mingyi Yang.

**Methodology:** Mingyi Yang, Muqi Zhou, Conglin Zhang.

**Resources:** Conglin Zhang.

**Supervision:** Conglin Zhang.

**Writing – original draft:** Mingyi Yang, Muqi Zhou.

**Writing – review & editing:** Mingyi Yang, Conglin Zhang.

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
