## [Decision Letter · Decision Letter 0]

26 Oct 2023

PONE-D-23-24928Effectiveness Evaluation of China's Water Resource Tax Reform Pilot and Path Optimization from the Perspective of Policy FieldPLOS ONE

Dear Dr. Zhang,

Thank you for submitting your manuscript to PLOS ONE. After careful consideration, we feel that it has merit but does not fully meet PLOS ONE’s publication criteria as it currently stands. Therefore, we invite you to submit a revised version of the manuscript that addresses the points raised during the review process.

Dear Authors,

Although the model of this manuscript is interesting, this manuscript needs to be revised very carefully. Please revise your manuscript according to the comments of reviewers accordingly. I appreciate that if you respond to all comments of reviewers in detail.

We look forward to receiving your revised manuscript.

Kind regards,

Mahdi Moudi, PhD

Academic Editor

PLOS ONE

A clean copy of the edited manuscript (uploaded as the new *manuscript* file).

Reviewers' comments:

Reviewer's Responses to Questions

**Comments to the Author**

1. Is the manuscript technically sound, and do the data support the conclusions?

Reviewer #1: No

Reviewer #2: No

2. Has the statistical analysis been performed appropriately and rigorously? 

Reviewer #1: I Don't Know

Reviewer #2: No

3. Have the authors made all data underlying the findings in their manuscript fully available?

Reviewer #1: Yes

Reviewer #2: Yes

4. Is the manuscript presented in an intelligible fashion and written in standard English?

Reviewer #1: No

Reviewer #2: Yes

5. Review Comments to the Author

Reviewer #1: The article has an interesting topic, but in general and with the following reasons, in my opinion, the article is not acceptable for publication.

1. The article does not have page numbers and line numbers so that it can be easily referred to the commons.

2. Sections 3.2 and 3.4 have no reference.

3. Many parts of the article are not referenced.

4. The article is more like a scientific report than a scientific research paper and research work.

5. The results are only presented as a report and there are no tables or figures to present the results.

6. There is no clear comparison of the results with other researches.

7. The separation of the parts of the article is not done properly.

8. It is not clear whether statistical methods were used to analyze the results or not.

Reviewer #2: Dear Authors

The manuscript (PONE-D-23-24928), entitled "Effectiveness Evaluation of China's Water Resource Tax Reform Pilot and Path Optimization from the Perspective of Policy Field” has been carefully reviewed.

The novelty of the study is unclear. Also, it is unclear how to collect data and conduct analyses in this research. Until such problems are not solved, the manuscript may be rejected.

However, the manuscript is weak, and substantial revision needs to be performed.

There are some suggested comments to improve the quality of the manuscript which are summarized as follows:

(There are no line numbers in the text which makes referring hard. I tried to do it based on the PDF file that was uploaded to the journal platform).

1- The abstract needs to be rewritten again. It does not cover the methodology used in the research. The verbs should be written in the past form. In addition, the abstract section is too long and can be shortened. The sentences at the end of the abstract section are all imperative and it is better to write them as suggestions.

2- In the Introduction section: some of the sentences should be cited by appropriate references because they represent some facts about China.

3- The Literature review section is weak. The author should focus on similar studies that have been implemented before. For example, those that used fuzzy sets or experienced water resources tax.

4- At the end of the literature review and based on the answers to the previous comment, the authors should conclude the novelty of the current study.

5- Figure 1: This flowchart is very unclear, where are the inputs? Where is the output?

6- On page 9, the authors mentioned that this study was carried out on 1+9 areas as a case study in China, so, a section should be added to the manuscript that shows the location of these case study areas.

7- The fuzzy system and analysis are unclear in the manuscript. These processes should be explained in the manuscript as to how they were used.

8- The results section is very weak! There are no useful graphs or tables that show the results. The authors only explained their results in a long text. This section should be revised in order to understand the analysis process better.

9- The conclusion section should be shifted after the discussion section!

10- The conclusion section is too long and should be shortened. In addition, this is not a good conclusion and should be revised.

11- The results should be compared with other similar studies.

Overall, there are many weaknesses in the manuscript that should be revised and resubmitted again. In the resubmitted file, please add line numbers in order to review better.

Thank you.

6. PLOS authors have the option to publish the peer review history of their article (what does this mean?). If published, this will include your full peer review and any attached files.

Reviewer #1: No

Reviewer #2: No

---

## [Author Response · Author response to Decision Letter 0]

20 Dec 2023

Dear editors,

Thanks for providing us with this great opportunity to submit a revised version of our manuscript. We appreciate the detailed and constructive comments provided by the reviewers. We have carefully revised the manuscript by incorporating all the suggestions by the reviewers. 

We hope this revised manuscript has addressed your concerns, and look forward to hearing from you.

Sincerely,

First author: Mingyi YANG

Second author: Muqi ZHOU

Corresponding author: Conglin ZHANG

Reply to Reviewer #1

Comments: The article has an interesting topic, but in general and with the following reasons, in my opinion, the article is not acceptable for publication.

Response: We appreciate your recognition of this topic and your detailed feedback and hope that the explanation has fully addressed all of your concerns. In the remainder of this letter, we discuss each of your comments individually along with our corresponding responses. 

To facilitate this discussion, we first transcribe your comments in italics types and then present our responses to the comments.

Reply to Reviewer #2

Comments: The manuscript (PONE-D-23-24928), entitled "Effectiveness Evaluation of China's Water Resource Tax Reform Pilot and Path Optimization from the Perspective of Policy Field” has been carefully reviewed.

The novelty of the study is unclear. Also, it is unclear how to collect data and conduct analyses in this research. Until such problems are not solved, the manuscript may be rejected.

However, the manuscript is weak, and substantial revision needs to be performed.

There are some suggested comments to improve the quality of the manuscript which are summarized as follows.

Response: We appreciate your clear and detailed feedback and hope that the explanation has fully addressed all of your concerns. In the remainder of this letter, we discuss each of your comments individually along with our corresponding responses.

To facilitate this discussion, we first transcribe your comments in italics type and then present our responses to the comments.

---

## [Decision Letter · Decision Letter 1]

19 Jan 2024

PONE-D-23-24928R1Effectiveness Evaluation of China's Water Resource Tax Reform Pilot and Path Optimization from the Perspective of Policy FieldPLOS ONE

Dear Dr. Zhang,

Thank you for submitting your manuscript to PLOS ONE. After careful consideration, we feel that it has merit but does not fully meet PLOS ONE’s publication criteria as it currently stands. Therefore, we invite you to submit a revised version of the manuscript that addresses the points raised during the review process.

We look forward to receiving your revised manuscript.

Kind regards,

Mahdi Moudi, PhD

Academic Editor

PLOS ONE

Journal Requirements:

Reviewers' comments:

Reviewer's Responses to Questions

**Comments to the Author**

1. If the authors have adequately addressed your comments raised in a previous round of review and you feel that this manuscript is now acceptable for publication, you may indicate that here to bypass the “Comments to the Author” section, enter your conflict of interest statement in the “Confidential to Editor” section, and submit your "Accept" recommendation.

Reviewer #2: (No Response)

2. Is the manuscript technically sound, and do the data support the conclusions?

Reviewer #2: No

3. Has the statistical analysis been performed appropriately and rigorously? 

Reviewer #2: No

4. Have the authors made all data underlying the findings in their manuscript fully available?

Reviewer #2: Yes

5. Is the manuscript presented in an intelligible fashion and written in standard English?

Reviewer #2: Yes

6. Review Comments to the Author

Reviewer #2: Dear Authors,

Thank you for efforts, but some minor errors should be revised.

1- Fig. 1: Add Latitude and Longitude to the map.

2- Table 4: header and columns are not matched.

3- the conclusion section is still long!

4- the number of references are too much! remove unnecessary references.

Thank you.

7. PLOS authors have the option to publish the peer review history of their article (what does this mean?). If published, this will include your full peer review and any attached files.

Reviewer #2: No

---

## [Author Response · Author response to Decision Letter 1]

3 Feb 2024

Dear editors,

Thanks for providing us with this great opportunity to submit a revised version of our manuscript. We appreciate the detailed and constructive comments provided by the reviewers. We have carefully revised the manuscript by incorporating all the suggestions by the reviewers. 

We hope this revised manuscript has addressed your concerns, and look forward to hearing from you.

Sincerely,

First author: Mingyi YANG

Second author: Muqi ZHOU

Corresponding author: Conglin ZHANG

Encl. Responses to the comments from Reviewers.

Reply to Reviewer #2

We appreciate your comments on this article and hope that our response adequately addresses all of your concerns. In the remainder of this letter, we will respond to each of your comments and will revise them accordingly. 

To facilitate this discussion, we first transcribe your comments in italics types and then present our responses to the comments.

Comment 1: Fig. 1: Add Latitude and Longitude to the map.

Response 1: We apologize for the lack of normality in Figure 1 without latitude and longitude. We appreciate your comments and have added latitude and longitude to figure 1.

Comment 2: Table 4: header and columns are not matched.

Response 2: We appreciate your careful review. The original title of Table 4 was indeed problematic, and we have revised the formatting of Table 4 with reference to papers of the same type. In addition, to increase the match between the Conclusions section and the empirical operationalization steps, we moved Table 4 to section 4.2.

Comment 3: the conclusion section is still long.

Response 3: We appreciate your feedback and agree with your suggestions. We have streamlined the conclusion section, reducing what was originally over 600 words to around 400 words, to ensure that the conclusion section provides a high level summary of the entire research.

Comment 4: the number of references are too much! remove unnecessary references.

Response 4: We appreciate your feedback and agree with your suggestions. We reduced the number of references to 43, and specifically deleted seven references in the original paper, including [22], [24], [27], [28], [32], [37], and [42]. On this basis we have revised the corresponding parts in the original paper and reordered and recoded all the references. For details, please see 'Revised Manuscript with Track Changes'.

---

## [Decision Letter · Decision Letter 2]

14 Feb 2024

PONE-D-23-24928R2Effectiveness Evaluation of China's Water Resource Tax Reform Pilot and Path Optimization from the Perspective of Policy FieldPLOS ONE

Dear Dr. Zhang,

Thank you for submitting your manuscript to PLOS ONE. After careful consideration, we feel that it has merit but does not fully meet PLOS ONE’s publication criteria as it currently stands. Therefore, we invite you to submit a revised version of the manuscript that addresses the points raised during the review process.

It seems that you have changed the affiliation of one author. I appreciate that if you explain the reason about this case. 

We look forward to receiving your revised manuscript.

Kind regards,

Mahdi Moudi, PhD

Academic Editor

PLOS ONE

Journal Requirements:

Additional Editor Comments:

Dear authors,

It seems that you have changed the affiliation of one author. Would you please explain why?

Reviewers' comments:

Reviewer's Responses to Questions

**Comments to the Author**

1. If the authors have adequately addressed your comments raised in a previous round of review and you feel that this manuscript is now acceptable for publication, you may indicate that here to bypass the “Comments to the Author” section, enter your conflict of interest statement in the “Confidential to Editor” section, and submit your "Accept" recommendation.

Reviewer #2: All comments have been addressed

2. Is the manuscript technically sound, and do the data support the conclusions?

Reviewer #2: No

3. Has the statistical analysis been performed appropriately and rigorously? 

Reviewer #2: Yes

4. Have the authors made all data underlying the findings in their manuscript fully available?

Reviewer #2: Yes

5. Is the manuscript presented in an intelligible fashion and written in standard English?

Reviewer #2: Yes

6. Review Comments to the Author

Reviewer #2: Dear Authors,

Thank you for your efforts. The manuscript is now accepted.

Thank you for your efforts. The manuscript is now accepted.

Regards.

7. PLOS authors have the option to publish the peer review history of their article (what does this mean?). If published, this will include your full peer review and any attached files.

Reviewer #2: No

---

## [Author Response · Author response to Decision Letter 2]

7 Mar 2024

We appreciate your clear and detailed feedback and hope that the manuscript has fully addressed all of your concerns.Thank you very much for your valuable comments and continued guidance on this paper, which has played a vital role in improving the quality of our paper.

---

## [Editor Report · Decision Letter 3]

11 Mar 2024

Effectiveness Evaluation of China's Water Resource Tax Reform Pilot and Path Optimization from the Perspective of Policy Field

PONE-D-23-24928R3

Dear Dr. Zhang,

We’re pleased to inform you that your manuscript has been judged scientifically suitable for publication and will be formally accepted for publication once it meets all outstanding technical requirements.

Kind regards,

Mahdi Moudi, PhD

Academic Editor

PLOS ONE
---

## [Editor Report · Acceptance letter]

18 Mar 2024

PONE-D-23-24928R3 

PLOS ONE

Dear Dr. Zhang, 

I'm pleased to inform you that your manuscript has been deemed suitable for publication in PLOS ONE. Congratulations! Your manuscript is now being handed over to our production team.

Kind regards, 

on behalf of

Dr. Mahdi Moudi 

Academic Editor

PLOS ONE